# COVID-19 and Historic Influenza Vaccinations in the United States: A Comparative Analysis

**DOI:** 10.3390/vaccines9111284

**Published:** 2021-11-05

**Authors:** Pranav Mirpuri, Richard A. Rovin

**Affiliations:** 1Chicago Medical School, Rosalind Franklin University of Medicine and Science, North Chicago, IL 60064, USA; 2Department of Neurosurgery, Aurora Neuroscience and Innovation Institute, Milwaukee, WI 53215, USA; Richard.rovin@aah.org

**Keywords:** COVID-19 vaccination, vaccine hesitancy, SARS-CoV-2, coronavirus, demographic factors, socioeconomic status, political preference, race

## Abstract

The COVID-19 vaccination effort is a monumental global challenge. Recognizing and addressing the causes of vaccine hesitancy will improve vaccine uptake. The primary objective of this study was to compare the COVID-19 vaccination rates in US counties to historical vaccination rates for influenza in persons aged 65 and older. The secondary objective was to identify county-level demographic, socioeconomic, and political factors that influence vaccination rates. County level data were obtained from publicly available databases for comparison and to create predictive models. Overall, in US counties the COVID-19 vaccination rate exceeded influenza vaccination rates amongst those aged 65 or older (69.4.0% vs. 44%, *p* < 0.0001). 2690 (83.4%) of 3224 counties had vaccinated 50% or more of their 65 and older residents in the first seven months of the COVID-19 vaccination roll out. There were 467 (14.5%) of 3223 counties in which the influenza vaccination rate exceeded the COVID-19 vaccination rate. Most of these counties were in the Southern region, were considered politically “red” and had a significantly higher non-Hispanic Black resident population (14.4% vs. 8.2%, *p* < 0.0001). Interventions intended to improve uptake should account for nuances in vaccine access, confidence, and consider factual social media messaging, especially in vulnerable counties.

## 1. Introduction

The COVID-19 pandemic vaccination roll-out is an unprecedented public health endeavor. The pandemic has swept the United States with over 670,000 dead and 41.9 million cases as of 20 September 2021 [1]. In fact, these official numbers are likely to be vast undercounts by many estimates due to rampant underreporting of cases and deaths [2]. Thus, with lives immediately at risk, it is crucial to appreciate barriers to vaccine access and to appreciate the important role of vaccine hesitancy.

Mandated, seasonal, and epidemic vaccination programs already exist in the United States [3]. The seasonal influenza vaccination program is well established and has decades of safety and efficacy data. Therefore, county influenza vaccination rates can serve as a standardized baseline of vaccine uptake. By comparing COVID-19 vaccination rates to those of influenza prior to the COVID-19 pandemic, county-level characteristics that influence vaccine uptake and hesitancy can be identified.

Comparing other vaccination rates to those of influenza is not without precedent. Nicholls et al. found in their comparison of influenza pneumococcal disease and shingles that interventions targeted towards older people would be more effective if they were vaccine specific and also emphasized the risks and benefits of vaccination to the individual and the community [4]. Similarly, Kwok et al. conducted a study on the association between influenza vaccination uptake with COVID-19 vaccination intention in nurses during the early phase of the pandemic [5]. They found that fewer than two-thirds of nurses intended to receive the vaccine when available.

In this study, we analyzed the influenza vaccination rates during the 2018–2019 influenza season and the COVID-19 vaccination rates through the first seven months of its availability using publicly available datasets. We specifically focused on persons 65 years of age and older as they are most at risk from COVID-19 and influenza disease severity and mortality; however, the overall COVID-19 vaccination rate across all ages was also analyzed [6,7]. Secondarily, we assessed the impact of county demographic, socioeconomic and political variables on COVID-19 vaccination rates to potentially identify at-risk populations and predictors for vaccination rates.

While our study can only provide evidence for correlation, this preliminary evidence can meaningfully inform future vaccination interventions which are vital in the setting of an urgent and evolving pandemic.

## 2. Materials and Methods

### 2.1. Study Design and Outcome Measures

This was a secondary analysis of publicly available data obtained from the Centers for Disease Control and Prevention (CDC) and from the County Health Rankings and Roadmaps (CHR&R) program of the University of Wisconsin Population Health Institute. For a complete list of data sources and limitations of the county demographic variables, refer to Appendix A.

The primary objective of this study was to compare county vaccination rates for COVID-19 through the first seven months of eligibility and the 2018–2019 influenza season for residents aged 65 and older. The secondary objective was to determine how county-level characteristics differentially influenced COVID-19 and influenza vaccination rates. “Rate” was defined as the percent of a county’s population that is fully vaccinated for either COVID-19 or seasonal influenza.

We evaluated the following socioeconomic, political, and demographic variables:
Race-related variables: Percent of county population belonging to the following races: Nonhispanic White (NHW), Nonhispanic Black (NHB), and Hispanic.Median Income: A continuous variable in dollars.High School Completion Rate: Percent of county population who have completed high school. Literacy is a well-recognized mediator of health outcomes [8]. In the COVID-19 setting, it has been shown to predict susceptibility to misinformation [9].Vaccine Hesitancy Estimate: Percent of county population who are estimated to be vaccine hesitant. The CDC derived the hesitancy rates by first estimating hesitancy on a Public-Use Microdata Area level using the Census Bureau’s 2019 American Community Survey 1-year Public Use Microdata Sample (PUMA) and then on the granular county level through a PUMA-to-county crosswalk from the Missouri Census Data Center [10]. Vaccine hesitancy is complex and multifaceted; these survey-based estimates are subject to sampling error and nonresponse bias. However, since we use the hesitancy data in conjunction with actual vaccination rates and not to find geographic or temporal trends, the limitations of the dataset are somewhat mitigated.Social Vulnerability Index (SVI): The SVI is a multidimensional metric for a community’s ability to respond to adversity (like pandemic illness). It is created from census data and incorporates 15 factors representing socioeconomic status, household composition and disability, minority status and language, and housing type and transportation. An index value of 0 indicates little to no vulnerability, whereas a value of 1 indicates an extremely vulnerable county. The breadth of the measure makes it a uniquely powerful tool for comparison in our study. Several publications describe its use in identifying social vulnerabilities during disasters and have also qualitatively demonstrated its application [11,12].COVID-19 Vaccination Coverage Index (CVAC): The CVAC is a measure that estimates the level of concern for difficulties in widespread COVID-19 vaccination coverage relating to supply and demand. The modular index is based on five themes including historic undervaccination, sociodemographic barriers, resource-constrained health systems, health accessibility barriers and irregular care-seeking behaviors. These themes are based on indicators such as proportion of individuals without insurance coverage, the proportion of individuals without visits to the doctor for routine checkup and the proportion of Medicare beneficiaries receiving the pneumococcal vaccine. Based on level of concern, the CVAC Index is categorized as follows: Very Low (0.0–0.19), Low (0.20–0.39), Moderate (0.40–0.59), High (0.60–0.79), or Very High (0.80–1.0). Although this measure has not been fully validated yet and there are limitations to its application, by aggregating baseline community characteristics it offers an interesting snapshot into the supply- and demand-related problems that a county might face [13].Segregation indices: these indices look at how evenly residents are distributed across a county’s census tracts. We include both the Black vs. White index (segblk) and the non-White vs. White index (segnonwht). The residential segregation index ranges from 0 (complete integration) to 100 (highly segregated).Internet access: Percent of households in a county with a broadband internet contract.

### 2.2. Statistical Methods

Parametric and non-parametric methods were used to compare the COVID-19 and influenza vaccinated cohorts. Comparisons were made based on county demographic, socioeconomic, and political data. Trends are described using Pearson’s correlation. Multiple regression was used to identify predictors of vaccination rate. Statistical significance was *p* ≤ 0.05. 95% confidence intervals were provided where appropriate. Stata 15 (College Station, TX, USA) and R (Version 4.1.0) were used for statistical analysis.

## 3. Results

### 3.1. COVID-19 and Influenza Vaccination Rates

Data from 3224 US counties were available for analysis. For persons 65 and older, the median rate for COVID-19 vaccination was 69.4% (Interquartile Range (IQR), 57.3–78.2%) and 41.1% (IQR, 33–51.5%) for all eligible ages. The median county level influenza vaccination rate for the 2018 season was 44% (IQR, 37–50%).

We next determined the number of counties meeting a 50% or greater vaccination threshold. 83% of counties (*n* = 2690) achieved a COVID-19 vaccination percentage of 50% or greater for their 65 and older population. By contrast, when all eligible ages were included, only 16.5% of counties (*n* = 534) reached the 50% threshold. For the influenza vaccine, 31% of counties reached the 50% threshold for those 65 and older (Figure 1).

Overall, there were five states with influenza vaccination rates higher than COVID-19 rates: Colorado, Georgia, Hawaii, Virginia, West Virginia (Figure 2). Taken together, in the first 7 months of the COVID-19 vaccine roll-out, acceptance was very high amongst persons 65 and older.

### 3.2. County-Level Characteristics and Vaccination Rates

We examined the influence that county-level demographic, socioeconomic, regional, and political features had on vaccination rates. Using Pearson’s method, we found positive correlations between the rate of COVID-19 vaccination in the 65 and older population and the percent of NHW and Hispanic residents in a county (0.09, 0.13, respectively, *p* < 0.0001). By contrast, there was a negative correlation between COVID-19 vaccination rate and the percent of NHB residents in a county (−0.16, *p* < 0.0001) (Figure 3).

For each county, we compared the COVID-19 vaccination rate to the influenza vaccination rate. There were 467 (14.5%) counties with COVID-19 vaccination rates lower than influenza rates. We compared these “lowvax” counties to “highvax” counties for which the COVID-19 vaccination rates exceeded the influenza rates (*n* = 2756). Characteristics of the lowvax and highvax counties are presented in Table 1; highlights are summarized below.

Overall, the median COVID-19 vaccination rate in the lowvax counties was 28.5% and 71.2% in the highvax counties (*p* < 0.001).

The lowvax counties had a significantly lower percentage (median) of non-Hispanic White residents (80.5% vs. 90.3%, *p* < 0.001), and significantly higher percentage of non-Hispanic Black (5.72% vs. 2.04%, *p* < 0.001) and Hispanic (5.8% vs. 4.05%, *p* < 0.001) residents compared to highvax counties.

Not only is the percentage of White and non-White residents significantly different between the low- and high-vax counties, their geographic distribution within counties also differs. The segregation index for White vs. Black residents is significantly higher in highvax counties compared to lowvax counties: 34.84 vs. 18.06, *p* < 0.001. The segregation index for White vs. non-White residents is also higher in highvax counties: 29.07 vs. 18.81, *p* < 0.001. When we examined the correlation between segregation and COVID-19 vaccination rates, we found a strong negative correlation in lowvax counties for both White vs. Black and White vs. non-White indices: −0.35 and −0.42, *p* < 0.0001, respectively. By contrast, in highvax counties, there was a positive correlation between segregation indices and the COVID-19 vaccination rates: 0.22 and 0.28, *p* < 0.001, respectively.

The social vulnerability index (SVI) was not significantly different for the lowvax and highvax counties: 0.49 vs. 0.5, *p* = 0.85. However, there were significant differences in several of the components of the SVI. For example, high school completion rate and median income were both higher in highvax counties, while unemployment was lower (Table 1).

The lowvax counties were predominantly in the southern region of the US and largely “red”. In the Midwest, only 32 of 1055 counties were lowvax. All these counties were red. In the Northeast, 4 of 217 counties had a greater percentage of persons vaccinated for influenza than for COVID-19. All these counties were “blue”. In the South, there were 276 of 1421 counties in which influenza vaccination percentage exceeded COVID-19 vaccination. Of these, 55 were blue, 221 were red. In the West, 73 of 418 counties were lowvax—32 were blue and 41 were red (Figure 4).

There were differences in vaccination rates based on political affiliation. The mean COVID-19 vaccination rate for the 538 blue counties was 67.87% (95% CI [65.77–69.98%]), and it was 63.80%, (95% CI [63.08–64.52%]) for the 2574 red counties. This difference is statistically significant, *p* < 0.0001. Blue counties also had a greater mean influenza vaccination rate than red counties: 46.42% (95% CI [45.60–47.24%]) vs. 42.4% (95% CI [42.01–42.78%]), *p* < 0.0001.

Regarding Internet access, there were differences between low- and high-vax counties. The median percent of households with any type of broadband internet access was significantly lower in the lowvax compared to the highvax counties: 70% vs. 75%, *p* < 0.001. Overall, as the percent of households in a county with internet access increased, the county’s COVID-19 vaccination rate increased (Pearson correlation 0.21, *p* < 0.0001). Furthermore, as the percentage of households with internet access increased, the COVID-19 vaccination rate decreased in the lowvax counties (Pearson correlation −0.31, *p* < 0.0001) but increased in the highvax counties (Pearson correlation 0.4, *p* < 0.0001). This is distinctly different than the findings for influenza vaccination. With increasing internet access, influenza vaccination rates overall, and for lowvax and highvax counties increased (Pearson correlation 0.4, 0.39, 0.4, *p* < 0.0001, respectively) (Figure 5).

### 3.3. Predictors of COVID-19 and Influenza Vaccination Rates in the 65 Years and Older Population

Multiple regression was used to build predictive models. We first used the percent of residents 65 and older fully vaccinated for COVID-19 as the outcome variable. The predictive variables in Model 1 were the percent of NHW, NHB, and Hispanic residents in the county. In Model 2, the segregation indices were added. For Model 3, we added SVI. Model 4 included region and political affiliation. In Model 5, we added the CVAC index (Table 2). Models 1–4 were repeated for the outcome variable representing the percent of residents 65 and older fully vaccinated for influenza (Table 3).

After controlling for demographic, socioeconomic, and political variables, the greatest negative predictor for the percentage of residents fully vaccinated for COVID-19 was residing in a Southern county. In the fully adjusted model for influenza vaccination, the greatest negative predictor was the SVI: as it increased, the percent of county residents 65 and older fully vaccinated for influenza decreased (Figure 6).

## 4. Discussion

This study compared county level COVID-19 vaccination rates to seasonal influenza vaccination rates for the 65 and older population. In all but five states and a minority of counties, the COVID-19 vaccination rate exceeded the baseline influenza vaccination rate. In addition, far more counties reached the 50% vaccination threshold for the COVID-19 vaccination than for the influenza vaccine (Figure 1 and Figure 2). This suggests that acceptance of the COVID-19 vaccine was quite high in this population. This is not unexpected, as the push for and urgency of COVID-19 vaccinations has been unprecedented in size and scope, particularly in the 65 and older population [14].

We used US county seasonal influenza vaccination rates from the 2018–2019 season as a baseline to which we compared COVID-19 vaccination rates. We chose influenza vaccination as the comparator because of its long history, established infrastructure, and specific targeting of the 65 and older population. We found important differences between counties with a lower and higher COVID-19 vaccination rate than influenza. The lowvax counties were largely in the Southern United States, were politically “red”, had higher unemployment, lower median income, lower high school graduation rate, and larger populations of people of color.

Our data also show that for the 65 and older population, as the percent of White or Hispanic residents in a county increases, the percent of those fully vaccinated for COVID-19 increases. By contrast, an increasing population of Black residents in a county is associated with decreasing COVID-19 vaccination rates (Figure 3). At first glance, this seems attributable to the black population’s historic distrust in the medical establishment. There is literature to support this: Lin et al. found in their systematic review of COVID-19 receptivity that Hispanics reported higher or similar acceptance to White-Americans in contrast to Black-Americans, who reported heightened suspicion and lower confidence [15]. However, recent polling from the Kaiser Family Foundation shows that the percent of Whites, Blacks, and Hispanics who received at least one dose of a vaccine was essentially the same (71%, 70%, 73%, respectively). Also, a greater percentage of Whites (14%) say that they will definitely not be vaccinated compared to Blacks (11%) and Hispanics (9%) [16]. As low confidence in the vaccine, then, is not the sole driver of this disparity, additional factors like vaccine access must be considered.

To that point, we found segregation was higher in highvax counties. And, as the segregation index increased, COVID-19 vaccination rates increased in highvax counties but decreased in lowvax counties. This suggests that people of color, particularly in lowvax counties, have greater barriers to vaccine access. This further suggests a solution: understand the geographic and cultural challenges populations of color face in accessing the COVID-19 vaccine, then plan the locations of distribution sites equitably [17,18].

As alluded to above, vaccination success is mediated by vaccine hesitancy. Vaccine hesitancy is a complex psychosocial phenomenon [19]. As outlined by the 2015 SAGE Working Group, hesitancy consists of three factors: complacency, convenience and confidence [20]. Complacency exists when the perceived risk from the vaccine-preventable disease is low and therefore the vaccine is not regarded as important for personal health. Convenience is a function of availability, affordability, and the overall appeal of the vaccine. Confidence requires trust in health care professionals, science, and government [21]. In fact, vaccine hesitancy has been on the rise for several years, visible in crises such as the increased resistance to the MMR vaccine in Somali-Minnesotans due to a now discredited study linking vaccination to autism [22,23]. Indeed, The World Health Organization called vaccine hesitancy one of the 10 greatest global public health threats in 2019, even before the COVID-19 pandemic [24]. The literature also shows that hesitancy against one vaccine quickly transfers to others, condemning the COVID-19 vaccine to an uphill battle from the moment it received emergency use approval [25]. The COVID-19 vaccine has uniquely faced politicization and has been the subject of rampant misinformation [26]. The misinformation “infodemic” has obfuscated true data on the vaccine, instead playing to fears for safety, personal liberty, and governmental overreach.

Ye recently reported that the COVID-19 vaccination rate “is strongly associated with political partisanship”, with vaccination rates in Republican counties lower than those in Democratic counties [21]. Our data are consistent with this political divide. Blue counties have significantly higher COVID-19 and influenza vaccination rates than red counties. As in the Ye paper, our data do not inform a causal mechanism. However, given the consistency of this association, specific messaging based on political party may enhance vaccination rates.

In a discussion of US vaccine hesitancy, the critical position of refugee, immigrant, and migrant (RIM) communities must also be considered in US vaccine hesitancy. It has been reported that RIM communities are disproportionately affected by COVID-19 and require special consideration in an equitable vaccination program [27]. Unfortunately, there is a paucity of literature on the specific drivers of vaccine hesitancy in RIM communities, most likely because RIM communities are not homogenous and vary significantly in factors such as socioeconomic status and cultural beliefs. Thus, it is recommended that vaccination efforts are undertaken in close collaboration with local partners who understand unique community-level factors. This allows for communities to be engaged directly to facilitate access to vaccines, lessen fear or mistrust of authorities, alleviate transportation challenges, and mitigate other barriers to widespread vaccination. For example, the use of portable immunization records has been suggested for migrant farmworkers, as they frequently relocate to find work, making it difficult to ensure continuity of care [28]. Interventions to alleviate vaccine hesitancy in other vulnerable communities in the US (for example, bringing vaccines to the people, messaging from trustworthy sources) can be applied to the RIM community with appropriate consideration to language and culture.

Lastly, the role of internet access in vaccine hesitancy cannot be overstated. Our data analysis highlights a compelling association between internet access and vaccination rates. For COVID-19, in highvax counties, increasing internet access is associated with increased vaccination rates, while in lowvax counties, increasing Internet access is associated with decreased vaccination rates. By contrast, for influenza, in both low- and high- vax counties, increasing internet access is associated with increased vaccination rates (Figure 5). This suggests a differential intensity of COVID-19 vs. influenza misinformation; it also suggests a differential susceptibility to COVID-19 misinformation in lowvax counties. But why are lowvax counties especially vulnerable to internet misinformation? Roozenbeek et al. found that people receiving information via social media were especially susceptible to COVID-19 misinformation and were more likely to resist vaccination [9]. It turns out that lower literacy, lower income, politically right-leaning, and self-perceived minority individuals are more susceptible to misinformation [9,29,30]. These are all hallmarks of lowvax counties.

## 5. Conclusions

Although this study cannot determine causation, the data showed a convincing picture of the impact of demographic, socioeconomic, and political factors on COVID-19 vaccine acceptance. Interventions intended to improve acceptance should focus on increasing access to and availability of the vaccine through community engagement, particularly in communities of color and in RIM communities. Factual and thoughtful social media messaging targeting vulnerable (lowvax) counties should also be considered.

## Figures and Tables

**Figure 1 vaccines-09-01284-f001:**
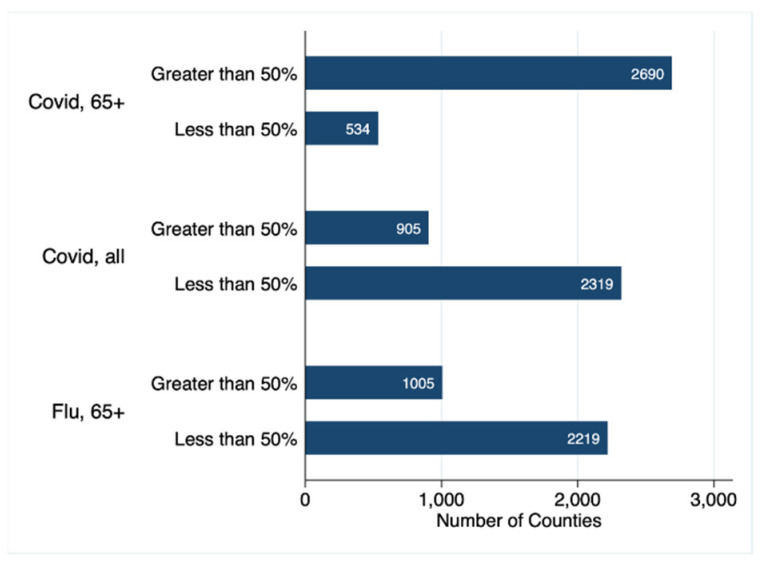
This bar graph shows the number of counties in the United States reaching the 50% vaccination threshold for COVID-19 vaccinations in the 65 and older population, COVID-19 vaccinations across all eligible ages, and for influenza vaccinations in the 65 and older population.

**Figure 2 vaccines-09-01284-f002:**
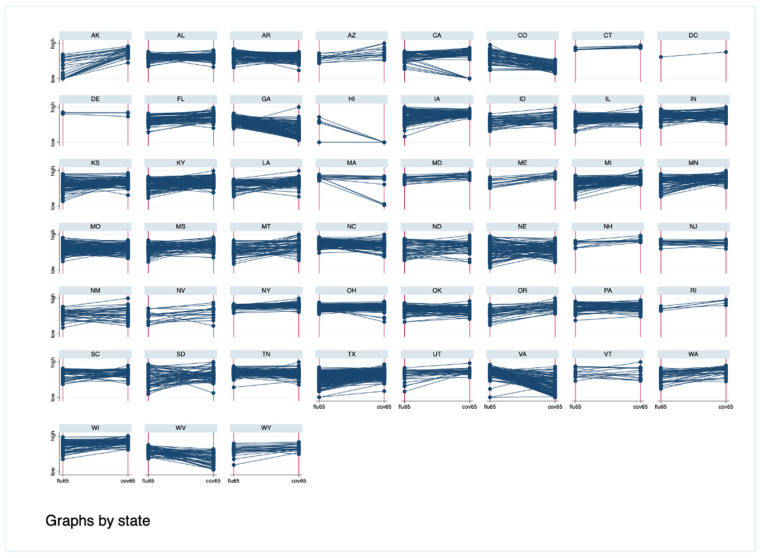
Slopegraphs comparing influenza vaccination rates (**left columns**) to COVID-19 vaccination rates (**right columns**). Each line represents a county within a state or territory.

**Figure 3 vaccines-09-01284-f003:**
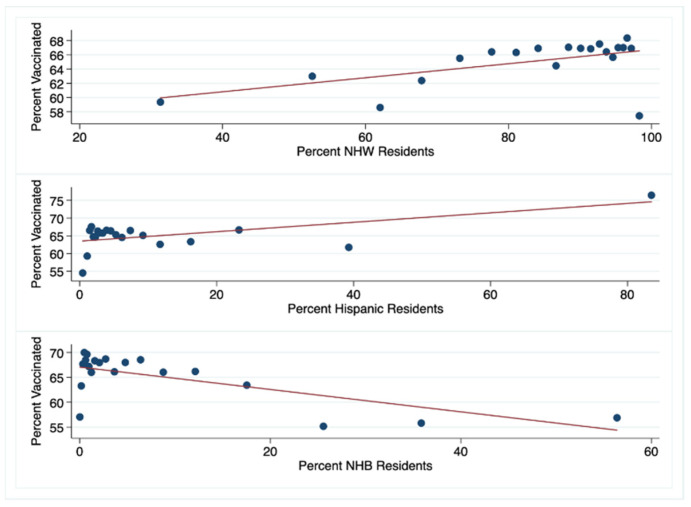
Binned scatter plots demonstrating the relationship between the rate of COVID-19 vaccination for the 65 and older population and the percent of county residents that are NHW, Hispanic, and NHB.

**Figure 4 vaccines-09-01284-f004:**
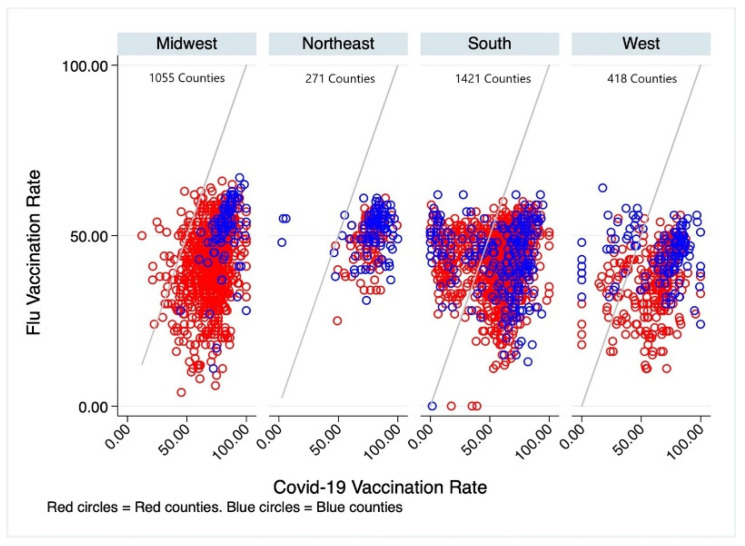
Influenza and COVID-19 vaccination in US regions and by political preference. The grey diagonal line represents equivalency of influenza and COVID-19 vaccination rate. Each circle represents a county. In counties above the diagonal line, the influenza vaccination rate exceeds theCOVID-19 vaccination rate. The counties below the diagonal line have COVID-19 vaccination rates that exceed influenza vaccination rates. Blue circles represent counties that voted for President Biden. Red circles represent counties that voted for President Trump.

**Figure 5 vaccines-09-01284-f005:**
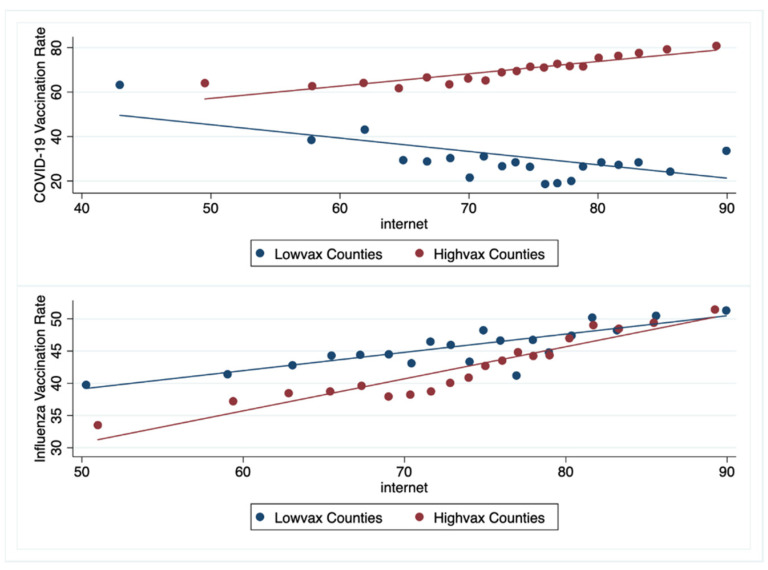
The association between Internet access and vaccination rates for low- and high- vax counties.

**Figure 6 vaccines-09-01284-f006:**
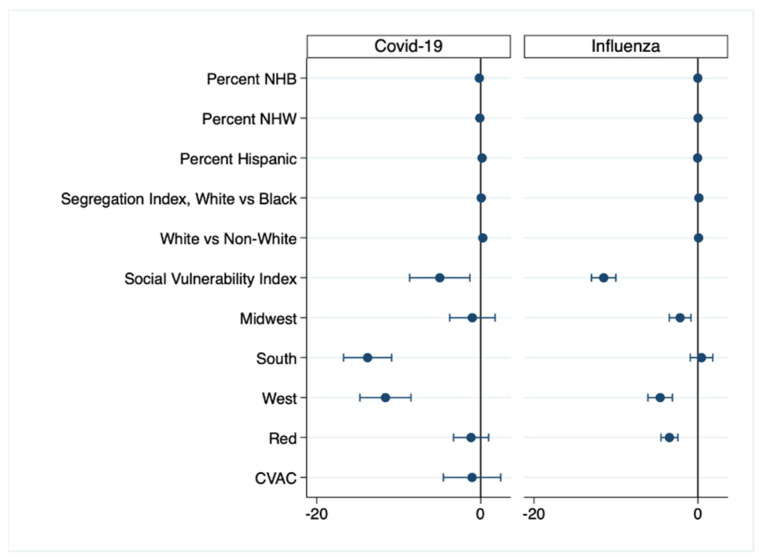
Coefficient plots for the fully adjusted multiple regression models of COVID-19 and Influenza vaccination rates. Horizontal bars represent 95% CIs. Left of the 0 line (negative numbers) indicates lower COVID-19 vaccination rates.

**Table 1 vaccines-09-01284-t001:** Comparison of counties with COVID-19 vaccination rates lower (Lowvax) and higher (Highvax) than their influenza vaccination rates.

	Total	Lowvax Counties	Highvax Counties	*p*-Value
	*n* = 3223	*n* = 467	*n* = 2756	
COVID-19 Vaccine, eligible ages	41.10 (33.00–51.50) *	19.10 (9.70–36.20)	42.60 (35.60–52.00)	<0.001
COVID-19 Vaccine, 65 and older	69.37 (57.37–78.20)	28.50 (13.70–45.20)	71.20 (62.30–79.10)	<0.001
Influenza Vaccine, 65 and older	44.00 (37.00–50.00)	46.00 (41.00–51.00)	44.00 (36.00–50.00)	<0.001
Percent NHB	2.37 (0.68–10.27)	5.72 (1.24–23.75)	2.04 (0.64–8.50)	<0.001
Percent NHW	89.24 (75.68–94.96)	80.49 (64.46–92.45)	90.26 (78.12–95.21)	<0.001
Percent Hispanic	4.23 (2.14–10.24)	5.80 (2.54–18.37)	4.05 (2.11–9.56)	<0.001
Segregation Index, White vs. Black	32.37 (0.00–50.29)	18.06 (0.00–36.52)	34.84 (0.00–51.74)	<0.001
Segregation Index, White vs. non-White	27.83 (15.91–37.47)	18.81 (0.00–30.75)	29.07 (18.50–38.33)	<0.001
Unemployment	3.17 (2.40–3.99)	3.30 (2.43–4.67)	3.16 (2.39–3.92)	<0.001
High School Completion	34.42 (29.67–39.15)	32.90 (28.00–38.38)	34.62 (29.93–39.28)	<0.001
Median Income (USD)	49,491 (41,865–57,333)	43,918 (35,096–55,439)	50,179.5 (42,835–57,478.5)	<0.001
Uninsured	9.04 (6.10–12.54)	9.74 (6.35–13.21)	8.93 (6.07–12.45)	0.17
Social Vulnerability Index	0.5 (0.25–0.75)	0.49 (0.24–0.8)	0.5 (0.25–0.74)	0.85
Internet Access	74 (68–79)	70 (61–78)	75 (69–79)	<0.001
Vaccine Hesitancy	0.13 (0.098–0.16)	0.099 (0.068–0.16)	0.13 (0.1–0.16)	<0.001
CVAC Index	0.5 (0.25–0.75)	0.48 (0.25–0.74)	0.5 (0.25–0.75)	0.73

* Median (IQR).

**Table 2 vaccines-09-01284-t002:** Multiple regression models to predict percent of county residents fully vaccinated for COVID-19.

	Model 1	Model 2	Model 3	Model 4	Model 5
Percent NHB	−0.262 ***	−0.278 ***	−0.303 ***	−0.171 ***	−0.169 ***
Percent NHW	−0.051	−0.013	−0.147 ***	−0.106 *	−0.106 **
Percent Hispanic	0.114 ***	0.192 ***	0.111 ***	0.170 ***	0.173 ***
Segregation Index					
White vs. Black		0.041 *	0.064 ***	0.070 ***	0.070 ***
White vs. non-White		0.304 ***	0.327 ***	0.259 ***	0.257 ***
Social Vulnerability Index			−13.592 ***	−5.609 ***	−4.995 **
Region (Index = Northeast)					
Midwest				−1.161	−1.022
South				−14.025 ***	−13.806 ***
West				−11.783 ***	−11.624 ***
Red County (Index = Blue)				−1.283	−1.179
CVAC Index					−1.055
Constant	70.289 ***	57.146 ***	74.423 ***	76.304 ***	76.346 ***
*n*	3224	3224	3142	3111	3111
R-squared	0.0415	0.1121	0.1372	0.2079	0.208
aR-squared	0.0406	0.1107	0.1355	0.2053	0.2052
Degrees of Freedom	3220	3218	3135	3100	3099

* *p* < 0.05, ** *p* < 0.01, *** *p* < 0.001.

**Table 3 vaccines-09-01284-t003:** Multiple regression models to predict county influenza vaccination rates.

	Model 1	Model 2	Model 3	Model 4
Percent NHB	0.140 ***	0.111 ***	0.173 ***	−0.01
Percent NHW	0.160 ***	0.177 ***	0.119 ***	0.021
Percent Hispanic	−0.073 ***	−0.074 ***	0.007	−0.038 **
Segregation Index				
White vs. Black		0.135 ***	0.137 ***	0.123 ***
White vs. non-White		0.075 ***	0.099 ***	0.079 ***
Social Vulnerabiliity Index			−11.712 ***	−11.499 ***
Region (Index = Northeast)				
Midwest				−2.175 **
South				0.435
West				−4.606 ***
Red Counties (Index = Blue)				−3.478 ***
Constant	28.966 ***	21.785 ***	30.367 ***	45.707 ***
*n*	3142	3142	3142	3111
R-squared	0.0412	0.1948	0.2562	0.2735
aR-squared	0.0403	0.1936	0.2548	0.2711
Degrees of Freedom	3138	3136	3135	3100

** *p* < 0.01, *** *p* < 0.001.

## Data Availability

The master dataset for this study has been archived in figshare https://figshare.com/ (accessed on 22 October 2021).

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
