# Peer review of "COVID-19 and Historic Influenza Vaccinations in the United States: A Comparative Analysis"

_vaccines, 2021, doi:10.3390/vaccines9111284_

Round 1
Reviewer 1 Report
Dear Authors,
I have read your interesting and informative proposal titled "COVID-19 and Historic Influenza Vaccinations in the United States: A Comparative Analysis", in which you have painstakingly laid out an analysis of relevant factors determining vaccination rates and vulnerability to misinformation leading to vaccine hesitancy.
The article is certainly well-written and relies on solid methodology. It is readable and well-assembled, although it is quite uncommon, and not advisable, to use contracted forms in scientific papers (see "isn't", in lines 315, 327). Also uncommon is to be reading political remarks in research papers, but it is admittedly warranted by the nature of your analysis itself, which focused on state/county characteristics, including political leanings.
The one aspect which you have failed to take into account is how vaccine hesitancy and lower access overall has impacted refugees, immigrant and migrant communities. That segment can hardly be overlooked, since it accounts for over 10% of the US population.
Maybe you need to do a bit more digging into the underlying determinants of vaccine hesitation/refusal among residents born abroad, including illegal aliens, by taking into the pictures a few comparisons with other countries as well, however briefly, and how different scenarios share similarities and differences. It would also be advisable to briefly mention some proposals as to how vaccine hesitancy can be tackled and minimized, particularly among vulnerable population segments, through effective and targeted information campaigns aimed at raising awareness and building trust.
Consider drawing upon the following sources in that regard:
Brandenberger, J.; Baauw, A.; Kruse, A.; Ritz, N. The Global COVID-19 Response Must Include Refugees and Migrants. Swiss Med Wkly 2020, 150, w20263, doi:10.4414/smw.2020.20263.
Cavaliere, A.F.; Zaami, S.; Pallottini, M.; Perelli, F.; Vidiri, A.; Marinelli, E.; Straface, G.; Signore, F.; Scambia, G.; Marchi, L. Flu and Tdap Maternal Immunization Hesitancy in Times of COVID-19: An Italian Survey on Multiethnic Sample. Vaccines 2021, 9, 1107, doi:10.3390/vaccines9101107.
Thomas, C.M.; Osterholm, M.T.; Stauffer, W.M. Critical Considerations for COVID-19 Vaccination of Refugees, Immigrants, and Migrants. The American Journal of Tropical Medicine and Hygiene 2021, 104, 433–435, doi:10.4269/ajtmh.20-1614.
Overall, I believe the article to be worthy of publication, provided that a few minor revisions are made meant to broaden the scope of the discussion and make it all more thorough and comprehensive.
Keep up the good work.
Reviewer 2 Report
Dear Editor and Authors,
I must say I read your manuscript titled “COVID-19 and Historic Influenza Vaccinations in the United 2 States: A Comparative Analysis” with quite an interest not only because it is well written and presented but mainly because it deals with an issue which, as I am sure you are aware is very poignant and relevant right now.
In this epidemiological study the two authors Dr. Mirpuri and Dr. Rovin performed an analysis of Covid-19 vaccination rates compared to previous years (2018-2019) Influenza vaccination rates in the over 65 population of the US.
This is a well conducted analysis with a solid methodological basis and good statistical method. Given that the data come from official (CDC, ect) sources we will have to assume their robustness. The manuscript is well written and presented and overall it requires only minor language editing if any.
One of the interesting findings, which is actually unfortunately expected for many well known socioeconomic and political reasons is that the rate of vaccination amongst black populations is lower that the comparable white and Hispanic ones.
The manuscript also makes an interesting political statement as there is, it seems a correlation of predominant political affiliation (democrat vs republicant) in the counties with higher and lower rates of Covid-19 vaccination respectively. This conclusion however I must caution the authors is “thin ice” and they must be very conservative with their conclusions (and this is coming from a European reviewer – albeit one who has lived in the past in the US for many years for study and work!). This is because, since they can not perform an individual survey of vaccination uptake and political affiliation to confirm that vaccination incidence is lower amongst republican voters the correlation they assert is tenuous at best.
Personally, I would omit the section as too politically charged, not only because I feel the purpose of science is to report the pure facts only and not make political assertions but also because their analysis and data show a correlation at best and not a definitive cause and effect.
In addition a variety of other well known and expected factors were shown to be correlated with Covid-19 vaccination rates such as internet access, segregation index, high school completion, median income and SVI.
Another of the limitations of the study I feel is the short period of Covid-19 they have assessed. Only the first 7 months of vaccine availability have been analyzed and I am not sure if this is truly representative as now a significant higher percentage has been vaccinated both because of the higher availability and diffusion of the vaccine and the vaccination effort and also the fact that initial vaccination rates may have been hindered from hesitancy and a “lets wait a bit” attitude.
However, I will agree with the authors in their assessment that vaccine hesitancy is not an exclusive race characteristic and the lower penetration of the vaccine in Southern, predominantly black counties and populations may reflect a lesser developed healthcare system in those areas and a lower vaccine availability. Basically, it is well known that healthcare delivery and access in many of these Southern economically depressed counties is lower than corresponding Northern ones!! Therefore, as previously mentioned I would be more keen to attribute the lower vaccination rates to this fact instead of their political affiliation!! To this effect lines 342 to 348, although true do not belong according to this reviewer in the discussion of a scientific paper but rather on a newspaper editorial.
Overall as I said I really liked this study/analysis. It is well conducted methodologically with interesting results and I would only try to make it less “political” and more scientific. In any rate, as the authors I am sure are aware “science is pure and the truth always reveals itself” therefore I suggest that they let the readers be the judge of the “messages” presented by their findings. I wish well to all and congratulations to the authors for a very poignant work.
